

# A hybrid anomaly detection method for high dimensional data

Xin Zhang[1], Pingping Wei[1] and Qingling Wang[2]

[1] School of Intelligent Science and Engineering, Yunnan Technology and Business University, Kunming, China
[2] Chongqing Technology and Business Institute, Chongqing, China

## ABSTRACT

Anomaly detection of high-dimensional data is a challenge because the sparsity of the data distribution caused by high dimensionality hardly provides rich information distinguishing anomalous instances from normal instances. To address this, this article proposes an anomaly detection method combining an autoencoder and a sparse weighted least squares-support vector machine. First, the autoencoder is used to extract those low-dimensional features of high-dimensional data, thus reducing the dimension and the complexity of the searching space. Then, in the low-dimensional feature space obtained by the autoencoder, the sparse weighted least squares-support vector machine separates anomalous and normal features. Finally, the learned class labels to be used to distinguish normal instances and abnormal instances are outputed, thus achieving anomaly detection of high-dimensional data. The experiment results on real high-dimensional datasets show that the proposed method wins over competing methods in terms of anomaly detection ability. For high-dimensional data, using deep methods can reconstruct the layered feature space, which is beneficial for gaining those advanced anomaly detection results.

## INTRODUCTION

Anomaly detection is an important part of data mining. There are many cases regarding anomaly appearance, such as system failure, abnormal behavior, data outliers, *etc.* Anomalies (aka outliers) have properties: (i) rarity, *i.e.,* it is difficult to label anomalies because anomaly instances are sparse; and (ii) type diversity, *i.e.,* there are many types of anomalies, such as, point anomalies, group anomalies, conditional anomalies, *etc.*

Anomalous features are more likely to be manifested in a low-dimensional space, but they are more hidden in a high-dimensional space. Due to high dimensionality, the distance contrast between data becomes similar (*Yu & Chen, 2019*; *Menon & Kalyani, 2019*), whereas, most anomaly detection methods explicitly or implicitly rely on distance contrast (*Li, Lv & Yi, 2020*). Obviously, high dimensionality easily leads to fail in anomaly detection for these methods relying on distance contrast. Furthermore, the data in a high-dimensional space presents a sparse distribution, which difficult affords rich information distinguishing abnormal and normal instances (*Soleimani & Miller, 2016*). In this case, anomaly detection for the data in a high-dimensional space is a challenge.

Corresponding author
Xin Zhang, jsjgcxz@163.com

Currently, anomaly detection methods allow to be divided into the following categories: (i) Distance metric-based methods do not acquire data distribution, *e.g.*, K-nearest neighbor (K-NN) (*Chehreghani, 2016*), Random Distances (*Wang et al., 2020b*). Distance metrics become more and more similar along with the increasing of data dimensionality, so such methods easily suffer negative effects of high dimensionality. (ii) Deep learning-based methods can not only learn deep features of the data, but also interpret the learned features (*Yuan et al., 2018*), *e.g.*, Bayesian Variational Autoencoder (BVAE) (*Daxberger & Hernández-Lobato, 2019*), and these methods in *Grosnit et al. (2022)*, *Bourached et al. (2022)*, *Grathwohl et al. (2019)*, *Goodfellow, Bengio & Courville (2019)* and *Ian et al. (2014)*. Such detection methods have unsupervised detection methods and supervised detection methods, where unsupervised detection methods do not rely on data labels but they are very sensitive to noise and missing data (*Parchami et al., 2017*), *e.g.*, Deep One-class Classification (DOC) (*Ruff et al., 2018*), Generative Adversarial Network (GAN) (*Li et al., 2019*), *etc*. The objective function of unsupervised detection methods is more used for data dimensionality reduction or data compression, so anomaly detection accuracy may be usually lower than that of supervised detection methods. Unlike unsupervised detection methods, Supervised detection methods show better detection performance because of relying on data labels, such as these methods in *Metzen et al. (2017)* and *Grosse et al. (2017)*. Unfortunately, labeling the data is a challenge if the amount of data is large or the data is multi-dimensional. Therefore, supervised anomaly detection methods are difficult to adapt to anomaly detection of high-dimensional data and large-scale data. (iii) Deep hybrid-based methods, such as Deep Neural Networks-Support Vector Machine (DNN-SVM) (*Inoue et al., 2017*), deep autoencoder and ensemble k-nearest neighbor (DAE-KNN) (*Song et al., 2017*), consist of deep methods and traditional detection methods; therefore, they inherit the characteristics of deep detection methods and traditional detection methods. Meanwhile, they own natural advantages in anomaly detection, but there needs trade-off computational cost and calculation accuracy. A typical representative of (iv) traditional detection methods are the support vector machine (SVM)-based methods, such as OC-SVM (*Ergen, Hassan Mirza & Serdar Kozat, 2017*), SVM (*Erfanin et al., 2016*; *Shi & Zhang, 2021*). SVM-based methods are susceptible to the linear inseparability of high-dimensional data (*Jerónimo Pasadas et al., 2020*); therefore, *Wang et al. (2020a)* proposed the improved SVM-based method.

The motivation of this article is to detect anomalies of high-dimensional data. Here, this article proposes a hybrid approach combining an autoencoder and a sparse weighted least squares support vector machine, namely AE-SWLS-SVM. Firstly, the autoencoder is used to extract low-dimensional features of high-dimensional data, thereby reducing data dimensionality and the complexity of the searching space. Then, in the low-dimensional feature space obtained by the autoencoder, the sparse weighted least squares support vector machine separates normal and abnormal features. Finally, the class labels being used to distinguish normal instances and abnormal instances are sent out, thereby realizing anomaly detection of high-dimensional data.

We summarize main contributions of this works.

(I) The proposed AE-SWLS-SVM can adapted well to high-dimensional environments during anomaly detection. Since the autoencoder captures the layered features from high-dimensional data, which provides beneficial environments for the sparse weighted least squares-SVM to distinguish the normal features and abnormal features.

(II) For high-dimensional data, the layered feature space reconstructed by deep methods is beneficial for gaining those advanced anomaly detection results. The contrast of distance between the data becomes difficulty as data dimensionality increases in high-dimensional spaces, however, in the layered feature space reconstructed by deep methods, the contrast of distance between the data becomes significant.

## MATERIALS AND METHODS

### Overall scheme

Figure 1 displays the overall scheme of the proposed method, which includes feature extraction, feature separation and instance reconstruction. In the first stage, namely feature extraction, the encoder captures low-dimensional features from the input data, which provides good environments for feature separation in the next stage. In the second stage, *i.e.,* feature separation, the sparse weighted least squares-support vector machine achieves the separation of abnormal and normal features in the low-dimensional feature space. In the third stage, namely instance reconstruction, the decoder reconstructs normal and abnormal instances from the separated normal and abnormal features. Finally, the learned class labels are output.

### Feature extraction

Autoencoders show excellent ability in capturing low-dimensional features of high-dimensional data. For simplicity, $\Re^h \xrightarrow{\Omega} \Re^l$ denotes that the autoencoder extracts low-dimensional features from high-dimensional data, where $\Re^h$ represents an $h$-dimensional high-dimensional space, and the data dimension in the space is $h$ dimensionality. Similarly, $\Re^l$ represents the corresponding low-dimensional feature space, and the dimensionality is $l$, and $l < h$. $\Omega$ represents our autoencoder, which consists of an input layer, an output layer and multiple hidden layers. The formal description for $\Omega$ is as follow.

Input layer $In_\Omega$ achieves the mapping of input data, *i.e.,* the data in $\Re^h$ is mapped onto $In_\Omega$.

Multiple hidden layers. The encoder and the decoder contain hidden layers, respectively, so we need to describe them, respectively.

(i) Hidden layers in the encoder. The input and the output of the $n$-th hidden layer in the $i$-th iteration are denoted as $E_{n,i}^{\text{in}}, E_{n,i}^{\text{out}}$, respectively. $E_{n,i}^{\text{out}}, E_{n,i}^{\text{in}}$ are calculated by Eqs. (1) and (2).

$$E_n^{\text{out}} = \nabla_n^e (w_n^i E_{n,i}^{\text{in}} + b_n^i) \tag{1}$$

$$E_{n,i}^{\text{in}} = E_{n-1,i}^{\text{out}} \tag{2}$$
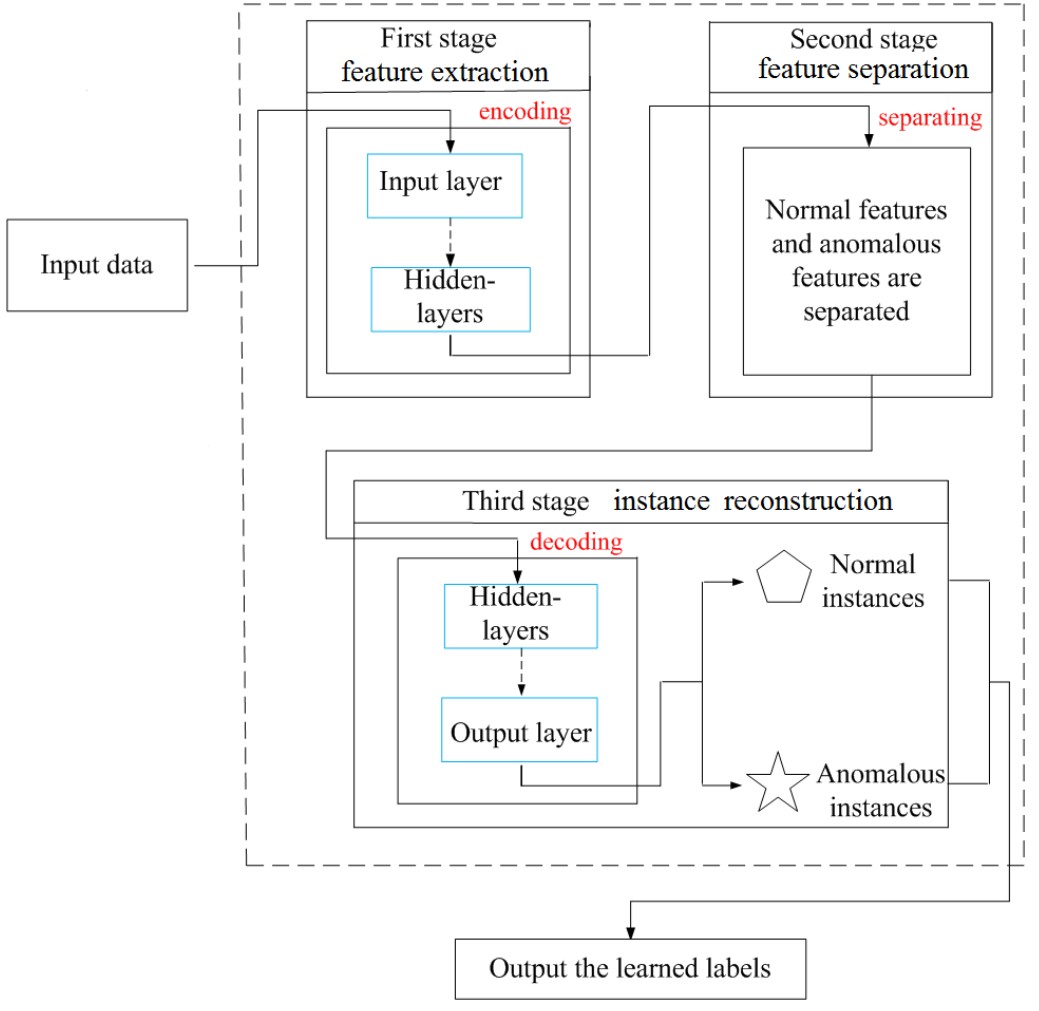

**Figure 1  Overall scheme.**

(ii) Hidden layers in the decoder. Correspondingly, the input and the output of the $m$-th hidden layer in the $i$-th iteration are denoted as $D_{m,i}^{\text{in}}, D_m^{\text{out}}$, respectively, as following,

$$D_m^{\text{out}} = \Delta_m^d (w_m^i D_{m,i}^{\text{in}} + b_m^i) \tag{3}$$

$$D_{m,i}^{\text{in}} = D_{m-1,i}^{\text{out}} \tag{4}$$

where $\nabla_n^e, \Delta_m^d$ are activation function in coding hidden layers and decoding hidden layers, respectively. $\mathbf{w}$ and $\mathbf{b}$ weight in hidden layers and bias.

Output layers $\text{out}_\Omega$ sends out the reconstructed instances. $\Omega$ isformally defined as follow

$$x \underset{\text{input}}{\rightarrow} In_\Omega \xrightarrow{E_{n,i}^{\text{in}}...E_{n,i}^{\text{out}},D_{m,i}^{\text{in}}...,D_m^{\text{out}}} \text{out}_\Omega \underset{\text{output}}{\rightarrow} z \tag{5}$$

where $x$, $z$ are the input and the reconstructed input, respectively. The loss function $L_\Omega$ of $\Omega$ is given in Eq. (6)

$$L_\Omega = ||x - z||^2 \tag{6}$$

Indeed, proper regularization can improve the ability of autoencoders to capture features (*Lu et al., 2017*), therefore, $L_\Omega$ is regularized by introducing regularization item and J–S(Jensen–Shannon) divergence $JS_{\text{sparse}}$, having that

$$L_\Omega = ||x - z||^2 + L_2 + JS_{\text{sparse}} \tag{7}$$

where $L_2$ (*Olshausen & Field, 1997*) is the regularization item and optimizes the weight of $\Omega$ in order to ensure that the components of **w** are as balanced as possible. $JS_{\text{sparse}}$ divergence is a variant based on K–L divergence. Because $JS_{\text{sparse}}$ divergence is symmetric, it can solve the problem of K-L divergence asymmetry (*Cattai et al., 2021*; *Li et al., 2021*). $JS_{\text{sparse}}$ divergence calculation formula is as follows

$$\begin{cases} JS_{\text{sparse}}(P_1||P_2) = \dfrac{1}{2}[KL(P_1||\dfrac{P_1+P_2}{2}) + KL(P_2||\dfrac{P_1+P_2}{2})] \\ KL(P_1||\dfrac{P_1+P_2}{2}) = -\sum\limits_{x \in X} P_1(x)\log\dfrac{1}{P_1(x)} + \sum\limits_{x \in X} P_1(x)\log\dfrac{1}{P_2(x)} \\ KL(P_2||\dfrac{P_1+P_2}{2}) = -\sum\limits_{x \in X} P_2(x)\log\dfrac{1}{P_2(x)} + \sum\limits_{x \in X} P_2(x)\log\dfrac{1}{P_1(x)} \end{cases} \tag{8}$$

where $P_1$ represents the true distribution of the data. $P_2$ is the theoretical distribution of the data or an approximate distribution of $P_1$.

## Feature separation

Support Vector Machine (SVM) is often used for classification tasks because of excellent classification ability. Anomaly detection can be thought as a binary-classification of normal and abnormal classes. Based on this, we improved the structure of SVM, *i.e.*, sparse weighted least squares was implemented to SVM, denoted as SWLS-SVM.

Given the input sample $\{(\hat{x}_i, \hat{y}_i)|i=1,2,..,n\}$, $\hat{y}_i$ is the label of $\hat{x}_i$. SWLS-SVM is defined as following

$$\left.\begin{aligned} &\min \frac{1}{2}||w||^2 + \frac{1}{2}\lambda \sum_{i=1}^{n} \beta_i \xi_i^2 \\ &L_i = w^T \phi(\hat{x}_i, \hat{y}_i) + b + \xi_i \end{aligned}\right\} \tag{9}$$

where $w$ and $b$ are weight and bias. $\lambda$ is a regularization parameter. $\beta_i$ is weight coefficient. $\xi_i$ and $L_i$ are an error item and the error function, respectively. $\phi(\bullet)$ is a non-linear mapping function. The goal of SWL-SVM is to minimize the $\frac{1}{2}||w||^2 + \frac{1}{2}\lambda\sum_{i=1}^{n}\beta_i\xi_i^2$.

SWLS-SVM employs a structural risk model, including empirical risk $\frac{1}{2}\lambda\sum_{i=1}^{n}\beta_i\xi_i^2$ and the regularisation term $\frac{1}{2}||w||^2$. If the regularization parameter $\lambda$ is larger, then the empirical risk becomes more important, certainly, this easily lead to over-fitting. If the regularization parameter $\lambda$ is small, then the empirical risk becomes less important, so that the effect of anomalies on the function $L_i$ can be ignored.

Equation (9) is a convex quadratic programming problem, and the dual problem can be obtained by using Lagrange multipliers. The constrained objective function in Eq. (9) can be transformed into an unconstrained Lagrangian function, having that

$$F(w,b,\xi_i) = \frac{1}{2}||w||^2 + \frac{1}{2}\lambda \sum_{i=1}^{n}\beta_i\xi_i^2 - \sum_{i=1}^{n}\alpha_i(w^T\phi(\hat{x}_i,\hat{y}_i) + b + \xi_i - L_i) \tag{10}$$

where $\alpha_i > 0$ is the Lagrange multiplier. To minimize $F(w,b,\xi_i)$, we set the partial derivatives of $w$, $b$, $\xi_i$, $\alpha_i$ to zero, respectively.

$$\left.\begin{array}{l}\dfrac{\partial F(w,b,\xi_i)}{\partial w} = 0 \rightarrow w = \displaystyle\sum_{i=1}^{n}\alpha_i\phi(\hat{x}_i,\hat{y}_i) \\[2ex] \dfrac{\partial F(w,b,\xi_i)}{\partial b} = 0 \rightarrow \displaystyle\sum_{i=1}^{n}\alpha_i = 0 \\[2ex] \dfrac{\partial F(w,b,\xi_i)}{\partial \xi_i} = 0 \rightarrow \alpha_i = \lambda\beta_i\xi_i \\[2ex] \dfrac{\partial F(w,b,\xi_i)}{\partial \alpha_i} = 0 \rightarrow w^T\phi(\hat{x}_i,\hat{y}_i) + b + \xi_i - L_i = 0\end{array}\right\}. \tag{11}$$

The mapping function $\phi(\bullet)$ in Eq. (11) can be converted into a kernel function $\kappa(\bullet)$ according to the KKT(Karush-Kuhn–Tucker) (*Peng & Xu, 2013*) condition, having that

$$\kappa((\hat{x}_i,\hat{y}_i),(\hat{x}_j,\hat{y}_j)) = \phi(\hat{x}_i,\hat{y}_i)^T\phi(\hat{x}_j,\hat{y}_j) \tag{12}$$

Let Eq. (12) be taken into Eq. (11), the linear Eq. (13) can be obtained by eliminating the variables $w$ and $\xi_i$, as follows

$$\begin{bmatrix} 0 & 1 & \vdots & 1 \\ 1 & \kappa((\hat{x}_1,\hat{y}_1),(\hat{x}_1,\hat{y}_1)) + 1/(\lambda\beta_1) & \vdots & \kappa((\hat{x}_1,\hat{y}_1),(\hat{x}_n,\hat{y}_n)) \\ \vdots & \vdots & \vdots & \vdots \\ 1 & \kappa((\hat{x}_n,\hat{y}_n),(\hat{x}_1,\hat{y}_1)) & \vdots & \kappa((\hat{x}_n,\hat{y}_n),(\hat{x}_n,\hat{y}_n)) + 1/(\lambda\beta_n) \end{bmatrix} \bullet \begin{bmatrix} b \\ \alpha_1 \\ \vdots \\ \alpha_n \end{bmatrix} = \begin{bmatrix} 0 \\ L_1 \\ \vdots \\ L_n \end{bmatrix} \tag{13}$$

By solving Eq. (13), the bias $b$ and the Lagrange multiplier $\alpha_i$ can be obtained.

$$f(\hat{x},\hat{y}) = \sum_{i=1}^{n}\alpha_i\kappa((\hat{x},\hat{y}),(\hat{x}_i,\hat{y}_i)) + b \tag{14}$$

Any semi-positive definite symmetric function can be used as a kernel function (*Jayasumana et al., 2014*), therefore, we obtain a positive definite function through calculating the cumulative distribution function (*Jayasumana et al., 2013*), having

$$\kappa(x_1,x_2) = (1 - (1-x_1^A)^B, 1 - (1-x_2^A)^B|A,B) \tag{15}$$

where $A$, $B$ are the non-negative kernel parameters.

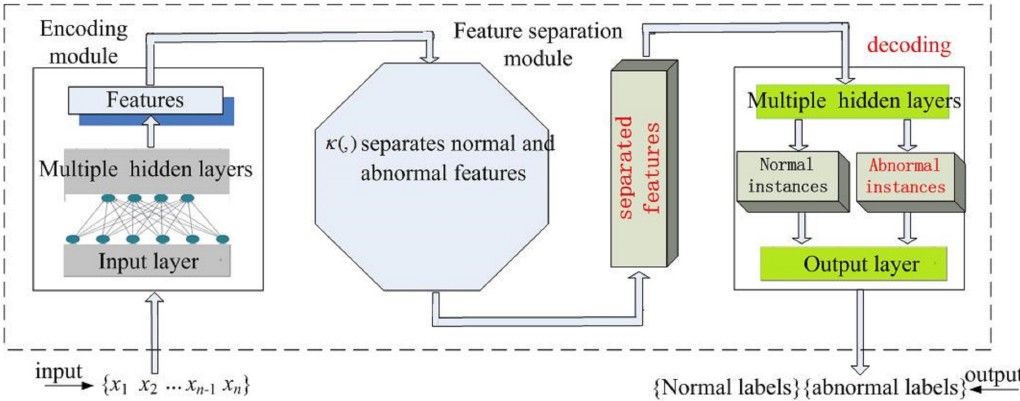

**Figure 2  Model structure.**

# MODEL

## Model structure and training

The proposed AE-SWLS-SVM consists of encoding module, feature separation module and decoding module, as shown in Fig. 2. The role of each module is described below. Module 1, *i.e.,* encoding module. In the module, hidden layers capture the low-dimensional features of the input data. The error of capturing low-dimensional features can be minimized by Eq. (7).

Module 2, *i.e.,* feature separation module. The module separates normal and abnormal features. In the low-dimensional feature space, the kernel function in Eq. (15) completes the separation of normal features and abnormal features.

Module 3, *i.e.,* decoding module. According to the separated features of the output, hidden layers in the module reconstruct normal and abnormal instances. Finally, the input layer outputs the learned normal and abnormal class labels.

The objective function of AE-SWLS-SVM consists of the loss function $L_\Omega$ of the autoencoder in Eq. (7) and the error function $f(\hat{x}, \hat{y})$ in Eq. (14) of SWLS-SVM, having that

$$T_{\text{target}} = \min\{L_\Omega + f(\hat{x}, \hat{y})\}. \tag{16}$$

The performance of AE-SWLS-SVM is related to parameters, so parameters of the model need to be cross-validated in advance, as shown in Algorithm 1. Firstly, the training set is used to train the model, and then the testing set is used to validate the trained model. By analyzing the testing results, the optimal kernel parameter value and the number of neurons are selected. The procedure in Step 2 to Step 18 mainly performs cross-validation of the number of neurons to obtain the optimal number of neurons $opt(\delta)$, where the procedure between Step 3 and Step 14 is the cross-validation of the kernel parameter to obtain the optimal kernel parameter value $opt(\gamma)$.

After obtaining the optimal parameters, we use the training set to train AE-SWLS-SVM, as shown in Algorithm 2. The process of Step 1 to Step 9 realizes the training of the

model. During the training, the objective function is iteratively calculated, when the model converges, we stop the training of the model and save the current training accuracy. From Step 10 to Step 14, we select the maximum training accuracy in the $t_{max}$-th training as the final output accuracy of the model, and then save the model for the $t_{max}$-th training.

Algorithm 1. Cross-validation of parameters.

Input: iteration epoch $T$, the number of neurons $\delta_1, \delta_2$, constant $\Delta\delta$, kernel parameter sets $\gamma_k$, training set Train_set, testing set Test_set.

Output:

Optimal kernel parameter value $_{opt(\gamma)}$, optimal number of neurons $opt(\delta)$.

Begin

| | |
|---|---|
| 1 | **for** $t = 1$ **to** $T$ **do**: |
| 2 | **for** $_{\delta=\delta_1}$ **to** $_{\delta_2}$ **do**: |
| 3 | **foreach** $\gamma$ **in** $\gamma_k$: |
| 4 | Train model SWLS-SVM(Train_set, $\delta, \gamma$); |
| 5 | **for** $i = 1$ **to** $I$ **do**: |
| 6 | Calculate the weight coefficient $\beta_i$; |
| 7 | Learn objective function $T_{\text{target}}$; |
| 8 | Calculate training accuracy $Train\_Acc =$ SWLS-SVM(Train_set, $\delta, \gamma$) ; |
| 9 | **end for** |
| 10 | Test model SWLS-SVM(Test_set, $\delta, \gamma$) ; |
| 11 | Calculate testing accuracy $Test\_Acc =$ SWLS-SVM(Test_set, $\delta, \gamma$); |
| 12 | **end foreach** |
| 13 | Select the optimal $\gamma$ so that maximize testing accuracy $\gamma(\max) = $ argmax($Test\_Acc(\gamma_{\max})$); |
| 14 | Obtain optimal kernel parameter $_{opt(\gamma)=\gamma(\max)}$; |
| 15 | $\delta = \delta + \Delta\delta$; |
| 16 | **end for** |
| 17 | Select the optimal $\delta$ so that maximize testing accuracy $\delta(\max) = $ argmax($Test\_Acc(\delta_{\max}, \gamma_{\max})$); |
| 18 | Obtain the optimal number of neurons $opt(\delta) = \delta(\max)$; |
| 19 | **end for** |

End

Algorithm 2. Model training.

Input: iteration epoch $T$, $_{opt(\gamma)}, opt(\delta)$, training set Training_set.

Output: training accuracy *Training_acc*.

Begin

1      **for** $t = 1$ **to** $T$ **do**:

2      Train model SWLS-SVM(Training_set; $_{opt(\gamma)}$; $opt(\delta)$) ; style=padding-left:1pc;,

3      **for** $i = 1$ **to** $I$ **do**:

4      Calculate the weight coefficient $\beta_i$;

5      Learn objective function $T_{target}$;

6      Calculate training accuracy *Training_acc* $(t) =$ SWLS-SVM(Training_set; $_{opt(\gamma)}$) ;

7      Save the $t$-th training accuracy $T\_acc\ (\text{save}(t)) = Training\_acc\ (t)$ ;

8      **end for**

9      **end for**

10      traverse the saved training accuracy $T\_acc\ (\text{save}(t))$ ;

11      Select the maximum training accuracy in the $t_{max}$-th training

12      $t_{max} = \arg \max(T\_acc\ (\text{save}(t)))$;

13      Save the model in the $t_{max}$-th training $_{Save=SWLS\text{-}SVM(Training\_set;opt(\gamma);opt(\delta);t_{max})}$;

14      Output training accuracy in the $t_{max}$-th *Training_acc* $= Training\_acc\ (t_{max})$;

End

## Model parameters

In order to better train AE-SWLS-SVM, the parameters that have a significant impact on the training results are investigated.

(1) Kernel parameter. To induce the best performance of AE-SWLS-SVM, the optimal value of kernel parameter is searched within a certain range.

(2) Weight coefficient $\beta_i$ is calculated as follows,

$$\beta_i = \begin{cases} 1, |\frac{\xi_i}{\bar{s}}|| \geq 1 \\ |\frac{\xi_i}{\bar{s}}|| \frac{\xi_i}{\bar{s}}| < 1 \\ 10^{-7} |\frac{\xi_i}{\bar{s}}| = 0 \end{cases} \tag{17}$$

where $\bar{s}$ isthe standard deviation of $\xi_i.|\xi_i/\bar{s}|$ indicates the absolute value of the error rate.

(3) Activation function. Activation functions used in machine learning are usually Sigmoid, tanh, ELU, ReLU, etc. The output of Sigmoid is only 0 and 1, which is suitable for the judgment of normal instances and abnormal instances. Therefore, Sigmoid is used as the activation function for the proposed AE-SWLS-SVM.

(4) Optimizer and learning rate. Adam is used as an optimizer for AE-SWLS-SVM. Adam not only possesses AdaGrad's ability to handle sparse gradients, but also has RMSProp's ability to handle non-stationary targets (*Kingma & Lei Ba, 2015*). Moreover, Adam can provide different adaptive learning rates for different hyper parameters.

(5) Number of neurons. The number of neurons was determined by cross-validation. Given that data dimension and data volume of the input data, using a certain range to configure the number of neurons can improve the ability of the model to resist over-fitting.

(6) Iteration epoch. During the training of AE-SWLS-SVM, we observe the change of training accuracy and dynamically adjust iteration epoch until the model can converge, and then stop training.

## EXPERIMENTS

### Datasets

In practical applications, real high-dimensional anomaly datasets are difficult to be obtained. Therefore, we selected real high-dimensional datasets of being often used for classification, and then used the method (*Campos et al., 2016*) to convert these high-dimensional datasets into anomaly detection datasets. We considered seven high-dimensional classification datasets (*i.e.*, U1–U7, the dimensions are greater than 165) to test the anomaly detection ability of the model. Then, we randomly selected five high-dimensional datasets from U1–U7 as the training set to train our model, and the selected process was repeated five times independently. In addition, three benchmark datasets (namely B1, B2, B3) were also selected, after our model is well trained, benchmark dataset B1, B2, B3 are used for parametric cross-validation and model structure validation. Table 1 gives a detailed description of these 10 datasets (seven high-dimensional datasets and three benchmark datasets).

### Assessment metrics

Accuracy and F1-score are used as evaluation indicators, and the calculation formula is as follows

$$\text{Accuracy} = \frac{\text{TP} + \text{TN}}{\text{TP} + \text{FP} + \text{TN} + \text{FN}} \tag{18}$$

$$\text{F1-score} = \frac{2\text{TP}}{2\text{TP} + \text{FP} + \text{FN}} \tag{19}$$

where TP represents the number of correctly predicted anomalous instances. TN represents the number of correctly predicted normal instances. FP represents the number of normal instances that are predicted to be anomalous instances. FN represents the number of anomalous instances that are predicted to be normal instances.

### Comparison methods

The proposed method is a hybrid method based on deep network architectures, so deep hybrid-based methods, *i.e.*, DNN-SVM (*Inoue et al., 2017*), DAE-KNN (*Song et al., 2017*), and deep networks-based methods, *i.e.*, GAN (*Li et al., 2019*), are used for comparisons. In addition, distance metric-based methods, *i.e.*, K-NN (*Chehreghani, 2016*) are also considered.

As for the proposed method, the adjustment range of the kernel parameter is defined as $\gamma_k = \{0.1, 0.3, 0.5, 1, 2, 3, 5\}$, the number of neurons $\delta = [\delta_1, \delta_2]$, $\delta_1 = 10$, $\delta_2 = 110$, and step $\Delta\delta = 20$. To have a fair comparison, the comparison method adopts the parameters observed in the corresponding literature.

**Table 1  Dataset information.**

| # | Benchmark datasets | Description (normal *vs.* anomaly) | Data volume | | Anomaly ratio | Data dimension |
|---|---|---|---|---|---|---|
| | | | Normal | Anomaly | | |
| B1 | Shuttle | Class '1' vs. Others | 1,000 | 13 | 1.28% | 9 |
| B2 | PenDigits | Other vs. Class '4' | 9,868 | 20 | 0.20% | 16 |
| B3 | Waveform | Others vs. Class '0' | 3,343 | 100 | 2.9% | 21 |

| # | High-dimensional datasets | Description (normal *vs.* outliers) | Data volume | | Anomaly ratio | Data dimension |
|---|---|---|---|---|---|---|
| | | | Normal | Anomaly | | |
| U1 | Arcene | Normal patterns vs cancer | 8,459,427 | 540,573 | 6.01% | 10,000 |
| U2 | Gisette | Zero vs non-zero values | 28,278,760 | 4,221,240 | 12.99% | 5,000 |
| U3 | Micro Mass | Zero vs non-zero values | 464,819 | 3,181 | 0.68% | 1,300 |
| U4 | Malware | Zero vs non-zero values | 2,894,954 | 37,772 | 1.29% | 1,087 |
| U5 | CNAE | Zero vs non-zero values | 918,537 | 7,023 | 0.76% | 857 |
| U6 | Epileptic Seizure | eizure vs. non-seizure | 11,321 | 179 | 1.56% | 179 |
| U7 | Musk | Musk vs non-musk | 79,699 | 269 | 0.34% | 168 |

We implemented corresponding algorithms of these methods using Python 3.8 on the Tensorflow 2.0 on Linux System. The environment setting are Intel i7 3.0 GHz CPU, and 32G memory. These algorithms run on the same GPU, and adopt the same configuration.

# RESULTS AND DISCUSSION

## Parameter testing of the model

AE-SWLS-SVM contains multiple hidden layers and is tested in the range {1, 2, 3, 5, 7, 10, 20, 30 }. The seven high-dimensional datasets are used as experiment datasets, *i.e.,* U1, U2, U3, U4, U5, U6, U7, we randomly selected five high-dimensional datasets from U1-U7 as the training set to train our AE-SWLS-SVM; meanwhile, the selected process was repeated five times independently. After our AE-SWLS-SVM is well trained, the benchmark dataset B1 is used as the testing set to test the number of hidden layers of AE-SWLS-SVM.

Testing results show that when the number of hidden layers reaches 3, the detection performance (including Accuracy and F1-score) of AE-SWLS-SVM tends to be stable, as shown in Fig. 3A. This shows that the proposed model is stable on the cases considered.

Let the number of hidden layers be equal to 3, then parameter are tested on the benchmark dataset B2 and B3, respectively, as shown in Figs. 3B and 3C. Results show that the detection performance of AE-SWLS-SVM is the best when kernel parameter $\gamma$ is equal to 0.5 and the number of neurons $\delta$ is equal to 50. Therefore, in the subsequent experiments, the parameters of AE-SWLS-SVM are configured as the number of hidden layers is 3, and $\gamma = 0.5, \delta = 50$.

## Performance comparison

Results on datasets U1-U7 show that AE-SWLS-SVM outperforms competitors DNN-SVM, DAE-KNN, GAN and K-NN in terms of anomaly detection performance on most datasets,

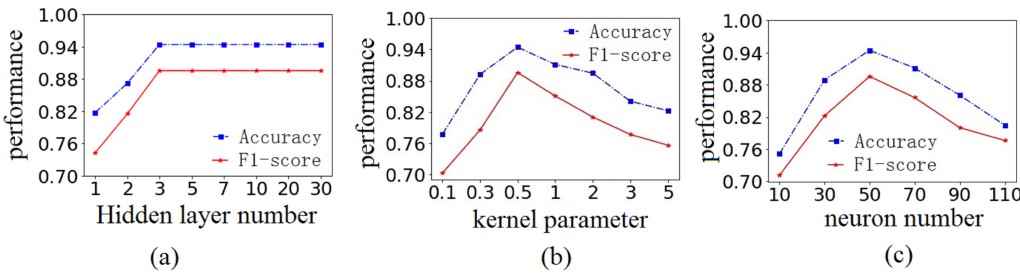

**Figure 3   Testing on benchmark dataset B1, B2 and B3.** Thesting results on benchmark dataset (A) B1, (B)2 and (C)3, respectively.

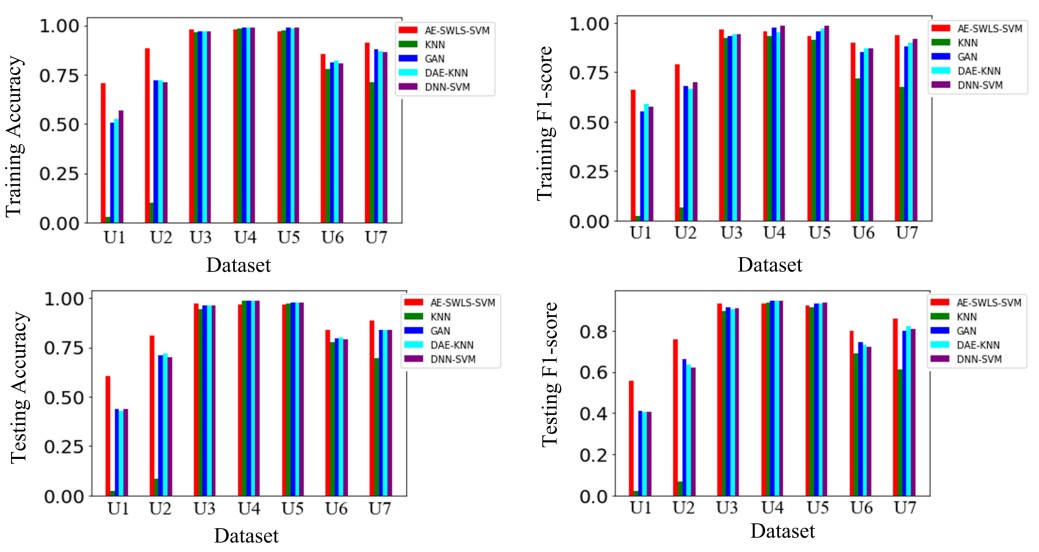

**Figure 4   Detected results.** These methods are compared in the Training/Testing Accuracy metric and Training/Testing F1-score metric.

as shown in Fig. 4. On higher dimensional datasets, such as U1 (dimension = 10,000) and U2 (dimension = 5,000), anomaly detection advantages of AE-SWLS-SVM are very significant. Unfortunately, the competitor K-NN almost fails on datasets U1 and U2. This is because the contrast of distance between the data becomes more difficulty as data dimensionality increases in high-dimensional spaces, unfortunately, the measurement manner based on distance may difficulty measures the attribute similarity of high-dimensional data. In fact, K-NN based on distance methods is prone to rely on measurement of distance. Although the detection results of these four competitors are better than our AE-SWLS-SVM on datasets U4 and U5, the advantages are not significant. Overall, AE-SWLS-SVM shows more advantages for anomaly detection on high-dimensional data.

## Discussion
### *Insights*

Compared with the above competitors, our model has outstanding advantages in terms of anomaly detection for the following reasons.

The autoencoder can capture the low-dimensional layered features from the input data, which is crucial because they provide a sufficient condition for the separation both anomaly features and normal features. The loss function of the autoencoder in Eq. (7) can minimize the error of the extracted low-dimensional layered features. This provides good environments for the kernel in Eq. (15) separating anomaly features from normal features. Through iteratively learning the objective function in Eq. (16), our model can gain good detection accuracy for anomalies.

For these anomaly detection methods, (i) distance metric-based detection methods, such as K-NN (*Chehreghani, 2016*), can work in low-dimensional spaces well, while their detection capabilities are restricted because the contrast between data points in high-dimensional spaces becomes similar, while (ii) deep learning-based detection methods, *e.g.*, GAN (*Li et al., 2019*), are suitable for working upon complex high-dimensional spaces due to owning nonlinear layers extracting important features or learning useful representations. Certainly, in terms of Generative Adversarial Networks (GANs), there exists an unavoidable mode collapsing so that the training is not easy. In regard to (iii) deep hybrid-based detection methods, such as DNN-SVM (*Inoue et al., 2017*), DAE-KNN (*Song et al., 2017*), the methods more and more popular since they inherit the advantages of deep network architectures and traditional detection methods. However, deep hybrid detection methods also show poor detection capabilities when traditional detection methods depend on data distribution or easily occur over-fitting, for example, DBN-Random Forest (*Kam Ho, 1995*) shows poor noise resistance and encounters high risk of over-fitting, due to random forest easily suffers over-fitting on the sample with relatively large noise (*Zheng & Zhao, 2020*; *Popolin Neto & Paulovich, 2021*).

### *Limitations*

The detection performance of the proposed method relies on the extracted features, which means that the quality of the extracted feature has important effects on the ability of the method. Additionally, due to lacking real anomaly datasets, the detection accuracies of most anomaly detection methods are restricted so that it is difficult to truly reflect the detection capabilities of them.

## CONCLUSION

This article proposes a hybrid method of combining an autoencoder and a sparse weighted least squares support vector machine for anomaly detection on high-dimensional data. The key thought is that the autoencoder extracts the low-dimensional layered features from high-dimensional data, in order to reduce the dimension of the data and the complexity of the searching space. In the low-dimensional feature space, the sparse weighted least squares-support vector machine separates anomalous features from normal features. Finally, the class labels of being used to distinguish normal instances and abnormal

instances are output, thereby completing anomaly detection of high-dimensional data. Results show that the proposed method is superior to competitors in terms of anomaly detection ability of high-dimensional data. In future works, we will look at addressing the issue of anomaly detection of noise interference. Noise can mask the rare anomalies so that anomalies are likely to be mistaken as noise.

### Funding
The authors received no funding for this work.

### Competing Interests
The authors declare there are no competing interests.

### Author Contributions
- Xin Zhang conceived and designed the experiments, analyzed the data, prepared figures and/or tables, authored or reviewed drafts of the article, and approved the final draft.
- Pingping Wei performed the computation work, authored or reviewed drafts of the article, and approved the final draft.
- Qingling Wang performed the experiments, analyzed the data, performed the computation work, authored or reviewed drafts of the article, and approved the final draft.

### Data Availability
The data and code is available in the Supplemental Files.

All datasets used in this work can be found at the UCI Machine Learning Repository: http://archive.ics.uci.edu/ml/.

### Supplemental Information
Supplemental information for this article can be found online at http://dx.doi.org/10.7717/peerj-cs.1199#supplemental-information.

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
