# Peer review of "A hybrid anomaly detection method for high dimensional data"

_PeerJ Computer Science, doi:10.7717/peerj-cs.1199_

## Round 0.1 · original submission · Major Revisions

Please consider the comments of the three reviewers and amend your manuscript according to them.

Reviewer 1 ·

Basic reporting

Introduction gives a short brief about machine learning methods and some study about Autoencoder related methods. I recommend you to improve this section by giving information about the most similar studies to your work. I suggest you to define the similarities and differences clearly.
Methods section gives details of the mathematical background for the study. But the references of these methods are not given (For example in line 146 'KKT condition').
In this section, there are some problems in mathematical notations:
In line 109 there is a Chinese character before the phrase 'are activation function in coding hidden layers'. Please give the definition.
In formula the definitons of D are not given.
In line 83, there is a confusion with Figure 1. You have expressed the stages as 'First stage,...etc.'. But in Figure one i am not able to understand which stage is related to which part of the Figure. I recommend you to add these stage information to Figure 1.
In line 99,100 there is some ambiguity problem in the sentence. I suggest you to check.
Algorithm 1 and Algorithm 2 are well-defined but there are some problems. In line 5 of Algorithm 1, the indice 'i' is not related to for each loops. This can also be seen in line 3 of Algorithm 2.
In formulas 17 and 18, Precision and Recall values are used. The definitions of these should be given.
In Section 4.3.1 in line 236, what is the amount of the data randomly selected as datasets? What is the repetitions given in 237? Repetitions of the with the same datasets and B1 or the randomly selection is repeated again. Please clarify.
In line 241, there is a phrase 'robust on the cases considered'. It is not clear. I recommend to give details in the current sentence instead of referencing the to the previous sentences for better understanding of the reader.
In Section 4.3.2, Line 252, i recommend to give an comment about why K-NN method fails in U1 and U2 datasets.

I discover some problems in English language so i recommend you a Proof check to ensure that other readers can easily read the paper.
Some suggestions where these corrections can be made are in lines : 44, 45 There is some semantic ambiguity.

Experimental design

This article presents a method to discover anomalies by using a hybrid method which combines Autoencoder archtitecture with a sparse weighted least squares SVM method.
Generally speaking, the purpose and necessity of the study is clearly defined. Although this methodology is an existing methodology where you can find studies by searching 'sparse weighted least squares SVM method' in Google Scholar, there is no finding that it is implemented as is in the proposal (i.e. in a ANN structure).

Validity of the findings

I suggest you to prepare a table which gives the train/test Accuracy, F1 and Kappa values for each method used for comparison in relation with Figure 4.

I expect a Data section in the study and give details about the datasets like, which amount is used as for training and validity (test). Why these datasets are selected. What is the expectation in anomaly rates in datasets for preference etc. Also please inquire data quality.

Underlying data have been provided in ‘http://archive.ics.uci.edu/ml/’ .It is checked. I can verify all datasets exist in this repository. CNAE with CNAE-9 name and Malware with ‘Detect Malware Types Data Set’ name. These should be corrrected. Authors should define how ‘Anomaly ratio’ in Table 1 is defined. How much of the dataset is used and the reason.

Reviewer 2 ·

Basic reporting

SUMMARY

The authors present a method for anomaly detection using an autoencoder in conjunction with a sparse weighted least squares support vector machine. There would appear to be some way to go before the empirical evaluation of the method is convincing and so I have the following points the authors may wish to consider:

MAJOR POINTS

1. It would be great if the authors could open-source their code on GitHub so that the empirical evaluation of the methods can be repeated.

2. It would be great if the precise contribution of the paper could be expanded on. From the point of view of the reviewer the idea of improving anomaly detection systems through learning a low-dimensional embedding is not novel and so the contribution would appear to be the particular type of model employed, namely the sparse weighted least squares support vector machine?

3. It does not seem appropriate to use 3 datasets for the purpose of cross-validation of the model architectures. How were these datasets selected? What about domain shift between datasets?

4. There are ideas in the use of variational autoencoders for the related problem of out-of-distribution detection [1,2,3] which are probably worth mentioning as sources of future ideas for the approach presented in this paper.

MINOR POINTS

1. At the moment it is difficult to distinguish section headings from the main text. It would be great if the authors could capitalise the section headings.

2. Is the x-axis of Figure 3b on a log scale? It would be great if this were explicitly stated. For clarity, it would be great if the caption of Figure 3 referred to the datasets by name.

3. In the introduction, it may be worth mentioning methods based on variational autoencoders and deep metric learning [4] which are capable of learning the embedding space according to class labels.

4. It would be great if there was consistent capitalisation of journal titles in the references.

5. Line 44: It would be great if this sentence could be revised.

6. In terms of the mathematical notation for the autoencoder, it would be great if the standard symbol for the d-dimensional space of the real numbers could be used i.e. \mathbb{R} in LaTeX. Additionally, I believe these h and l-dimensional spaces should not take arguments because they represent sets and not functions.

7. Line 98: Why is the input layer a function of the autoencoder represented by \Omega?

8. Line 109: There is an extraneous Chinese character.

9. It would be great if the authors could adopt standard notation for autoencoders from the machine learning literature e.g. [5]

10. Line 113: Typo in the subscript for the loss function.

11. Typo in Equation 6 for the subscript on the Jensen-Shannon divergence term.

12. Definition of Jensen-Shannon divergence should feature the theoretical and the empirical distribution of the data presumably?

13. Typo in subscripts of Equation 7.

14. The set X is never defined. Presumably it represents the input space?

15. Line 131: Indices should be placed on the set in subscript/superscript

16. The argument of the minimisation problem in Equation 8 is not specified.

17. Line 134: Why is the hat taken off the labels y?

18. Is it correct to refer to the regularisation term of the SVM as a confidence interval?

19. The matrix-vector multiplication in Equation 12 is incorrect?

20. The typesetting of Equation 16 makes the quantity look like a square root sign instead of a fraction.

21. Adam was accepted at ICLR 2015.

22. Line 195: ELU, ReLU

23. The citation for the GAN should be Goodfellow et al. [6].

24. It is not explained how the GAN is used explicitly for anomaly detection?

25. In the appendix why can't the caption for Table 1 be included on the same page as the table?

REFERENCES

[1] Daxberger and Hernández-Lobato, J.M., Bayesian variational autoencoders for unsupervised out-of-distribution detection, arXiv, 2019.

[2] Grathwohl et al., Your classifier is secretly an energy based model and you should treat it like one. In International Conference on Learning Representations, 2019.

[3] Bourached et al. Generative model‐enhanced human motion prediction. Applied AI Letters, 3(2), p.e63. 2022.

[4] Grosnit et al., High-Dimensional Bayesian Optimisation with Variational Autoencoders and Deep Metric Learning, arXiv, 2022.

[5] Goodfellow, Bengio, Courville, Deep Learning, 2019.

[6] Goodfellow, I et al. Generative adversarial nets. Advances in neural information processing systems, 27. 2014.

Experimental design

cf. basic reporting

Validity of the findings

cf. basic reporting

Additional comments

cf. basic reporting

·

Basic reporting

There are instances where the paper lacks clarity and the grammar could be improved. For example:

• “It is a challenge for anomaly detection to high-dimensional data, because the data upon a high-dimensional space is so sparsely distributed that hardly provides rich information to distinguish anomalous instances from normal instances.”
• line 83, “Fig.1 displays the overall scheme of the proposed method, including three stages, respectively, low-dimensional feature extraction, feature separation, and instance reconstruction.”
• line 99, “Hidden layers contain the coding hidden layer of capturing low-dimensional features and 100 decoding hidden layer of reconstructing the input, respectively, we describe them.”
• line 242 , ”Let the number of hidden layers be equal to 3, kernel parameter÷ and the number of neurons are verified in a similar way, and then tested them on the benchmark datasets B2 and B3, respectively, as shown in Figure 3(b) and Figure 3(c).”

There are other examples where the article would benefit from proofreading.

The proposed model addresses an existing problem outlined in the article that has been described in the introduction. The prior work has been described in sufficient detail and has been appropriately referenced.

The article follows the structure specified in the Standards Section policy with two exceptions:
• One section is labelled as “Methods” instead of “Materials and Methods”;
• There is no “Discussion” section.

A “Discussion” section would be a useful addition to account for the performance of the proposed model compared with that of its competitors, e.g. GAN or KNN. This section could also describe the implications of the results of the experiments as well as future directions. These points are not covered in other sections of the paper.

The figures are of sufficient resolution, are well-described and are labelled.

The raw data has been shared.

The work is self-contained, represents an appropriate unit of publication and contains all the results relevant to the hypothesis.

Some terms are not clearly defined or defined with respect to other terms which themselves are not defined, i.e. in line 134, under the ‘Feature Separation’ section, the definition of xi is cyclical. A conceptual or semantic definition that is independent of f(x) and L_i is required. Additionally, it would be useful to provide an intuitive explanation of L_i. For example, is L_i the loss function?

Experimental design

The article describes original primary research and fully complies with the aims and scope of the journal.

The research question is well-defined and it is clear how the proposed approach contributes to filling the knowledge gap. The authors address the problem of anomaly detection when the feature space is high dimensional and they propose the AE-SWLS-SVM model as a solution that differs from previous approaches.

The hyperparameters, e.g. the number of neutrons and the kernel parameter, appear to have been optimised on the test set. Convention states that a separate validation set be dedicated to hyperparameter optimisation and that the results be reported on the test set to prevent overly optimistic reporting.

Furthermore, it is unclear why low-dimensional datasets with fewer examples were employed for the hyperparameter optimisation and robustness testing. Would it not be better to perform all the experiments on the larger, high-dimensional datasets?

It is not obvious whether the performance comparison testing was conducted on the same data that was used for testing or whether each dataset was randomly split into train and test sets. If only the training results were reported there would be a high risk of overfitting. The experiments should be conducted on separate train and test sets.

While the authors do include ample numbers of mathematical formulae to support their descriptions, there are certain details that limit the reproducibility of the work. It is unclear from the description what is meant by “separation of features” with regards to the SVM when it is applied to the low-dimensional feature space. Does it assign class labels at this point? Or does it perform some other function? The figures seem to suggest that the labels are applied only once the inputs have been reconstructed. Intuitively, the best approach might have been to use the AE to merely reduce the dimensionality of the data while assigning labels with the SVM. Beyond training the encoder, what use do the decoder and reproduced inputs serve? If the AE is used purely for dimensionality reduction with the SVM performing the classification, this should be stated more clearly in the text and shown more explicitly in the figures.

Moreover, there is a confusing use of terminology on line 165: “reconstructed low-dimensional feature space”. There is either the low-dimensional feature space which is the output of the encoder or the reconstructed space which refers to the reconstructed inputs produced by the decoder. Presumably in this instance, the authors are referring to the low-dimensional feature space, as they are discussing the “feature separation” module.

Finally, it is unclear what the x^hat in f(x^hat, y) represents in equation (15). Does x^hat refer to the low-dimensional features?

Validity of the findings

Decisions do not seem to be guided by any subjective determination of impact, degree of advance, novelty or being of interest to only a niche audience.

The underlying data has been provided.

Too few datapoints were used for performing the hyperparameter search and evaluating robustness. One of the larger high-dimensional datasets would have been more appropriate.

The conclusions are well-stated and are directly related to the research questions. However, the aforementioned flaws of the experimental methodology limit their strength, i.e. the ambiguity surrounding the data separation leads to questions surrounding the validity of the final statement: “Experimental results show that the proposed method is superior to these mainstream detection methods in anomaly detection ability”.

Additional comments

It is unclear what the Chinese character on line 9 represents.

In line 102, as omega depicts dimensionality reduction, why is it used as an input to the encoder layers, E? Does E represent a functional in this case?

In equation (6), as omega already refers to the dimensionality reduction operation, a different letter should be used to represent to the regularisation term.

In the “Feature Extraction” subsection of “Methods”, the relationship between A and D should be made explicit, i.e. in the form of an equation.

The relationship between x (input), z (reconstructed input), E and D should be made explicit, i.e. an equation where x is fed into E^in_0 and where z is the output of D^out_M.

---

## Round 0.2 · accepted · Accept

Congratulations on the acceptance of your manuscript!

Reviewer 1 ·

Basic reporting

I have checked the corrections and additions which were completed by the authors according to my recommendations. I confirm these corrections/additions are complete and adequate for this section.

Experimental design

I have checked the corrections and additions which were completed by the authors according to my recommendations. I confirm these corrections/additions are complete and adequate for this section.

Validity of the findings

I have checked the corrections and additions which were completed by the authors according to my recommendations. I confirm these corrections/additions are complete and adequate for this section.

Reviewer 2 ·

Basic reporting

comments included below

Experimental design

comments included below

Validity of the findings

comments included below

Additional comments

The authors have adequately addressed all concerns and so I recommend acceptance.

·

Basic reporting

All the comments about the basic reporting were addressed.

Experimental design

All the comments about the experimental design were addressed.

Validity of the findings

All the comments about the validity of the findings were addressed.

Additional comments

All additional comments were addressed.